# The Impact of Mandatory Vaccination Law in Italy on MMR Coverage Rates in Two of the Largest Italian Regions (Emilia-Romagna and Sicily): An Effective Strategy to Contrast Vaccine Hesitancy

**DOI:** 10.3390/vaccines8010057

**Published:** 2020-01-30

**Authors:** Davide Gori, Claudio Costantino, Anna Odone, Beatrice Ricci, Magda Ialonardi, Carlo Signorelli, Francesco Vitale, Maria Pia Fantini

**Affiliations:** 1Department of Biomedical and Neuromotor Sciences, University of Bologna, 40138 Bologna, Italy; mariapia.fantini@unibo.it; 2Department of Health Promotion Sciences, Maternal and Infant Care, Internal Medicine and Medical Specialties (PROMISE), University of Palermo, 90127 Palermo, Italy; claudio.costantino01@unipa.it (C.C.); francesco.vitale@unipa.it (F.V.); 3School of Medicine, University Vita-Salute San Raffaele, 20132 Milan, Italy; odone.anna@hsr.it (A.O.); signorellicarlo2307@gmail.com (C.S.); 4School of Hygiene and Preventive Medicine, University of Bologna, 40138 Bologna, Italy; beatricericci@gmail.com (B.R.); magda.ialonardi@gmail.com (M.I.)

**Keywords:** vaccine hesitancy, mandatory vaccinations, coverage rates

## Abstract

**Background**: Vaccine hesitancy has increased worldwide, leading to reduction in vaccination coverage rates. In particular, reduction in the coverage for the trivalent Measles-Mumps-Rubella vaccine has led to an increase of measles cases. The aim of this study is to analyze the coverage rates for the MMR vaccine in the Emilia-Romagna Region (RER) and Sicily Region (SR) between 2009 and 2018, and to correlate any significant change to index events which could have modified the trend of vaccination rates. **Methods**: Official aggregate data on vaccination coverage at 24 months provided by the RER and the SR were analyzed through trend analysis and related to important index events. **Results**: The two regions showed similar results; both achieved the lowest coverage rates in 2015 and both showed an increase in the rates after the introduction of mandatory vaccinations for access to schools. In 2018, both reached the starting point before the decrease. **Conclusions**: Our results confirm the effectiveness of legislative coercive measures in favor of vaccination. A potential decrease in the coverage rates may be observed as a result of an attenuation of the positive effects of coercive measures over time. It is thus necessary to combine these measures together with information campaigns and political initiatives at different levels (i.e., national, regional).

## 1. Introduction

Vaccination is one of the greatest public health achievements of human history [1] and one of the most effective and safe interventions for primary prevention [2]. Reaching and maintaining high vaccination coverage (VC) may turn into eliminating or drastically reducing the incidence of preventable infectious diseases. Nevertheless, anti-vaccination movements have existed since the 18th century [3,4], and fear and mistrust have arisen every time a new vaccine has been introduced, leading to the modern concept of vaccine hesitancy (VH) [5], defined as the “delay in acceptance or refusal of vaccination despite availability of vaccination services [6]”. In recent years, a worldwide increase in VH and a reduction in coverage rates has been observed worldwide, so that the World Health Organization (WHO) has listed VH as one of its top 10 health threats facing the world in 2019 [7]. The decision not to vaccinate comes from a complex decision-making process influenced by numerous factors related to population, context, environment, historical period and single vaccine.

Measles, mumps and rubella (MMR) are three very dangerous infectious diseases which can cause significant morbidity and lead to potentially fatal illness, disability and death. Live attenuated vaccinations for MMR have been licensed in the market for almost 50 years [8].

Widespread vaccination against measles has prevented an estimated 21.1 million deaths worldwide between 2000 and 2017 [9]. The disease has been targeted for elimination in all six WHO regions and one of the aims of the Global Vaccine Action Plan 2015–2020 is to eliminate measles in at least five regions by 2020. In 2003, in Italy, the first national measles elimination plan was implemented and a two-dose schedule was introduced, starting with the 2002 birth cohort, with the first dose given at 12–15 months of age and the second dose at 5–6 years. In 2017, when MMR vaccination became mandatory for children up to the age of 16 years, VC at a national level was less than 92% for the first dose at 2 years of age [10].

Enforcing mandatory vaccinations is one of the strategies that has been considered in recent years by some countries (e.g., France and Germany) in order to face this issue for low VC. Many countries, like Italy, have decided to adopt this solution, also taking into account that mandatory vaccinations have always been controversial and related to opposition and disputation [11]. The elimination plan was updated in 2010 following these considerations. However, from 1 January 2017 to 31 December 2018, a large epidemic outbreak occurred in Italy (7437 measles cases notified to National Health Authorities), with an incidence of 52 cases per million inhabitants.

The Emilia-Romagna Region (RER), despite having one of the best coverage rates for the MMR vaccine in Italy, has experienced the VH phenomenon. In 2008, 180 cases of measles were recorded, going from a less than 0.5 to a 4.6 incidence rate per 100,000 inhabitants. Another epidemic outbreak occurred in 2010 (129 cases), 2011 (198 cases) and 2014 (208 cases). In those outbreaks, young non-vaccinated or single-dose vaccinated adults were affected in particular. In addition, 499 rubella cases were recorded in 2008, in all age groups [12]. All of the aforementioned events led regional authorities to enacting the Regional Law n.19 on November 25th 2016. Specifically, this law introduced mandatory vaccination in order to access educational and recreational services in public kindergartens. In Sicily, in 2018, 1111 measles cases (44% of the cases reported in Italy) were notified, making it the Italian region that also registered the highest incidence (222 cases per million).

In 2017 a new National Plan for Vaccine Prevention (NPVP) 2017–2019 was approved in Italy [13], followed by the National Law 119/2017, which increased the number of mandatory vaccinations from four to ten (vaccination for polio, diphtheria, tetanus, pertussis, hepatitis B, haemophilus influenzae B, measles, mumps, rubella and chickenpox), introducing fines for the “hesitant” and refusing parents [14,15].

The aim of this study is to analyze the coverage rates for the MMR vaccine at 24 months (first dose) and at 7 years (full vaccination cycle) of age in the Emilia-Romagna Region (RER) and the Sicily Region (SR), between 2009 and 2018, and to correlate any significant changes to index events.

## 2. Materials and Methods

This study investigates infant and childhood MMR immunization coverage rates and trends in the period 2009–2018 in RER and SR. Regional vaccination programs are scheduled as follows:RER: 1 dose at the 13th month, and the second dose at 6 years;SR: 1 dose at the 13–15th month, and the second at 5–6 years.

Vaccination coverages were reported for first dose at 24 months and for the full vaccination cycle at 7 years of age. Results are reported by year and major areas of RER and SR. Current immunization coverage data in RER and SR are expressed as the most updated coverage rates in the period 2009–2018 (cross-sectional design analysis). We considered vaccine coverage rates at 24 months (first dose) and at 7 years of age (two doses). Vaccination coverage was calculated using as numerator the number of children who underwent vaccination in the index year and as denominator the number of eligible resident children at the beginning of the year.

### 2.1. Data Collection in Emilia-Romagna Region

RER is a northeastern Italian region with 4,459,477 inhabitants, and is sixth for demographic density in Italy [16]. RER is divided into 8 Local Health Authorities (LHAs). The data on vaccination coverage were collected and reported for the following three macro-areas:Romagna: Local Health Authorities of Forlì-Cesena, Ravenna, Rimini;North Emilia: Local Heath Authorities of Piacenza, Parma, Reggio Emilia, Modena;Central Emilia: Local Health Authorities of Bologna, Imola, Ferrara.

Data on MMR vaccination coverage at 24 months and 7 years of age were provided by the Ministry of Health and RER in aggregate form and are available at the regional website [17].

### 2.2. Data Collection in Sicily Region

Sicily is a southern Italian region of 4,999,891 inhabitants, fourth for demographic density in Italy [16]. The Sicily region is divided into 9 Local Health Authorities (LHAs). Also, in this case, vaccination coverage data were reported for two main macro-areas, divided as follows:Western Sicily: Local Health Authorities of Agrigento, Caltanissetta, Palermo, Trapani;Eastern Sicily: Local Health Authorities of Catania, Enna, Messina, Siracusa, Ragusa.

Data on MMR vaccination coverage at 24 months and 7 years were provided by the Sicilian Health Department that record data into a Digital Regional Vaccine Registry (AVR). All the data collected by the Vaccine Registry of the 9 Sicilian LHAs during a year were checked within the 28th of February of the following year, verified and sent to the Regional Health Department in aggregate form.

### 2.3. Statistical Analysis

Immunization coverage trends over time were analyzed using crude rates of vaccine coverage over the study years. We used Joinpoint (JP) regression models to identify statistically significant trends and changes in trends (increasing/decreasing) in MMR vaccination during 2009–2018 in RER and 2012–2018 in SR. This method analyzes changes in data trends by connecting several different line segments, and identifies points where a statistically significant change over time in the linear slope of the trend occurs. In addition, an annual percentage change (APC) for each line segment is estimated. The APC is tested to determine if it is different from 0% [18]. JP Regression Programme version 4.7.0.0 was used to carry out the analysis.

The χ2 test was used to analyze differences in vaccine coverage between specific years characterized by important events. In particular, we decided to take into account any important event which occurred in Italy in the time frame considered that could have had any impact on the political decisions, media coverage and population perceptions (e.g., other infectious diseases outbreaks, laws, court sentences). The significance level was set to *p* < 0.05.

## 3. Results

### 3.1. Vaccine Coverage at 24 Months (First Dose)

As shown in Table 1, vaccine coverage at 24 months in RER was substantially stable between 2009 (93.5%) and 2012 (92.4%) and was the highest at national level^25^. It underwent a significant decrease from 2013 (91.1%), until it reached the historically lowest level in 2015 (87%), with a 5.4% decrease in just three years. From 2016 (87.2%), the regional trend showed an increase in vaccination coverage up to 93.5% in 2018. The vaccination coverage rates were constantly lower in the Romagna area during the observation period.

Vaccine coverage at 24 months in SR showed an increase between 2009 (86.8%) and 2011 (91.6%), followed by a significant decrease from 2013 (88.9%) onwards, until it reached the historically lowest level in 2015 (79.2%), with a 9.7% decrease in just three years. Since 2016 (81.1%), the vaccination coverage showed an increase up to 90.9% in 2018. In Eastern Sicily the vaccination coverage rates were the lowest.

### 3.2. Vaccine Coverage in Childhood (Second Dose)

The 7-year coverage rates for the second dose of MMR showed a stable increase until 2014 with values around 90%, never reaching the 95% target. Since 2015, instead, there was a decrease in coverage rates with two doses. Area differences were evident. Romagna showed the lowest vaccination coverage (Table 2). The 7-year coverage rates for the second dose of MMR in SR showed a stable increase until 2018, up to 84.7%. Eastern Sicily showed a lower vaccination coverage compared to the Western area of the region, where values higher than 90% were observed in 2017 (Table 2).

### 3.3. Joinpoint Analysis

#### 3.3.1. Emilia-Romagna Region

The trend of vaccine coverage for 100,000 residents at 24 months from 2009 to 2018 showed only one break point in 2015 (that was the year of lowest coverage in vaccination for both targeted infant populations). From 2009 to 2015 there was a significant reduction in vaccination coverage with an average annual 1.04% decrease (i.e., about 1040 24-month children less per 100,000 were vaccinated each year compared with the previous year). From 2015 to 2018 there was a non-significant 3.50% increase per year (i.e., about 3500 24-month children more per 100,000 were vaccinated each year compared with the previous year) (Figure 1). 

We analyzed differences in the proportion of vaccinated children between 2012 and the following years and between 2015 and the following years. The percentage of vaccinated children decreased significantly (*p* < 0.01) from the year 2012 to all subsequent years. However, the percentage of vaccinated children (χ2 = 32.34, *p* < 0.001) was significantly higher in 2018 compared with 2012. (Table 3). No significant difference in the percentage of vaccinated children was found in 2015 compared to 2016 (χ2 = 1.04, *p* = 0.31), whereas there was a significantly higher percentage of vaccinated children in 2017 (χ2 = 318.16, p<0.001) and 2018 (χ2 = 851.35, *p* < 0.001) compared with 2015 (Table 4).

#### 3.3.2. Sicily Region

The trend of vaccine coverage per 100,000 residents at 24 months from 2012 to 2018 showed only one break point in 2015, which is the year of lower coverage in vaccination for both targeted infant populations. From 2012 to 2015 there was a non-significant reduction in vaccination coverage with a 2.88% decrease per year (i.e., about 2880 24-month children less per 100,000 were vaccinated each year compared with the previous year). From 2015 to 2018 there was a non-significant 6.83% increase per year (i.e., about 6830 24-month children more per 100,000 were vaccinated every year compared with the previous year) (Figure 2). 

The percentage of vaccinated children in the year 2012 was significantly higher (p < 0.01) compared to all subsequent years. However, in 2018 the percentage of vaccinated children was significantly higher than in 2012 (χ2 = 95.92, p < 0.001) (Table 5). We observed a significantly higher percentage of vaccinated children in 2016, 2017 and 2018 compared with 2015 (Table 6). 

### 3.4. Index Events

For the purposes of this work we considered some index events which could have modified the trend of vaccination rates: 1) The Rimini Court sentence of 15 March 2012 [19] which supported the possible association between vaccine and autism [3]. 2) The withdrawal of adjuvant trivalent influenza vaccine lots from the Italian Medicines Agency (Agenzia Italiana del Farmaco, or AIFA), following the occurrence of two suspected deaths within 48 hours of vaccine administration in November 2014 (the first suspected death occurred in Eastern Sicily) [20]. 3) The reversal of the Rimini Court sentence by the Bologna Appeal Court in 2015 [21]. 4) A meningitis epidemic outbreak spread in the Tuscany region (next to RER) [22,23,24]. 5) 2016 ER Regional Law n.19, which introduced mandatory vaccination for access to educational and recreational services. 5) 2017 National Law, which increased the number of mandatory vaccinations for school attendance from four to ten.

## 4. Discussion 

We present comprehensive immunization coverage rates of two Italian regions, focusing on MMR vaccine programs for 24-month-old and 7-year-old children between 2009 and 2018. The trivalent Measles-Mumps-Rubella (MMR) vaccine has been one of the most targeted by hesitant parents due to possible (but never proven) adverse effects. Among European parents, misleading knowledge, beliefs, and perceptions on the MMR vaccine, and general negative attitudes and behaviors toward vaccinations, were significantly associated with lower MMR vaccination uptake rates [25]. This resulted in an increase of measles cases, particularly in Italy.

In particular, in Italy, the coverage rates have been decreasing since 2012 [26,27,28]. One of the possible causes of this steep decrease can be found in the Rimini Court sentence of 15 March 2012 [19]. This sentence (which obtained widespread media coverage) supported the possible association between vaccines and autism. This fact, along with other general causes like the role of social media (e.g., Facebook, Twitter, YouTube or Wikipedia) in a Web 2.0 context [29], caused increasing fear in the population and contributed to the development of no-vax movements in the RER [3] and generally at a national level. In addition, AIFA, due the occurrence of two suspected deaths within 48 hours of vaccine administration (the first suspected death occurred in Eastern Sicily) [20], decided to withdraw, as a precautionary measure, some lots of adjuvant trivalent influenza vaccine in November 2014. This event was again widely covered by media, which highlighted the potential negative consequences that the influenza vaccine could have on a population’s health. These two events may have been important contributors to enhancing the general suspicion that has developed in the Italian population on vaccination practice in the last years. This negative effect has led at a regional level (as shown in our results) to the achievement of the minimum historically recorded coverage rates in the year 2015 [30].

Concerning regional differences, from 2009 to 2018 in RER and from 2012 to 2018 in SR, lower MMR vaccination adherence rates were constantly observed respectively in Romagna and Western Sicily. In particular, these areas of RER and SR were involved in a greater activity of anti-vaccination associations, with a consequent increase of VH among the general population [31,32].

Notably, the Rimini Court sentence of 2012 was pronounced in Romagna and the first of two suspected deaths within 48 hours of the "Fluad^®^" vaccine administration in 2014, that opened investigations by the Judiciary, occurred in Western Sicily [33,34]. The reversal of the Rimini Court sentence by the Bologna Appeal Court in 2015 [21], along with an unexpected meningitis epidemic outbreak spread in the Tuscany region (which is next to RER and which had important media coverage throughout Italy) in the same years probably had an incremental impact on general vaccine coverage trends in hesitant parents [22,23,24,25,26,27,28,29,30,31,32,33,34,35].

The vaccination coverage at 24 months for the MMR vaccine in RER showed an important reduction between 2012 and 2015, leading regional and national health authorities to take urgent measures. In 2016 RER enacted the Regional Law n.19, which introduced mandatory vaccination for access to educational and recreational services for kindergarten-age children. The decision by RER was reinforced by the 2017 National Law 117/2017, which increased the number of mandatory vaccinations for primary school attendance from four to ten [36]. In addition to the law, the Italian Ministry of Health enacted on 18 February 2017 the 2017–2019 National Vaccination Prevention Plan (Piano Nazionale Prevenzione Vaccinale, or PNPV). This document recognizes, as a public health priority, the reduction and the elimination of the burden of infectious diseases preventable by vaccine, through the identification of effective and homogeneous strategies to be implemented throughout the national territory, underlying the fundamental role of vaccination as primary and efficient preventive measure for eradicating infectious diseases.

We think that the aforementioned events, along with the enactment of the two laws (at a Regional and National level) should be considered as the main causes of the trend reversal observed from 2015 onwards. Our data show the same increasing trends in the two regions. In fact, RER and SR achieved, in the year 2018, the promising result of reaching their starting point (despite their initial different coverage rates) before the steep decrease due to all the aforementioned negative events which occurred in Romagna and Western Sicily. These data and trends are in line with those observed at a national level in 2018, where differences in the percent coverage between northern and southern regions of Italy were reported [37].

Despite the latency between the legislative measures and vaccination coverage at 24 months, we observed a 4.1% increase in RER and a 6.4% increase in SR in vaccination coverage for MMR in the year 2017 following the RER Law 19/2016, and an additional 2.5% and 5.3% in SR in the year 2018 following the PNPV 2017–19 and especially Law 117/2017 deliberation. This confirms the effectiveness of legislative measures in favor of vaccination, together with information campaigns and political initiatives at different levels. It must be pointed out that the importance of the reversal of the initial court ruling not based on scientific theories, media campaigns supporting vaccinations, and the PNPV 2017–2019 may have contributed to the increase in coverage rates. 

As far as the limitations of the study are concerned, we have to underscore the possible heterogeneity between data extracted from RER and SR sources. In addition, regional differences may be due to late adherence to vaccination related to peculiar healthcare and educational services organization, and different vaccination rates may depend on socioeconomic disparities which are highlighted at the National level by the GDP (Gross Domestic Product) [38]. In relation to the GDP, a disparity in National BHCL (Basic Health Care Level) fulfillment, defined by the Ministry of Health, has been described previously [15]. These differences cannot be considered properly as a limitation of this work, but also as a consequence of the peculiarities in the Italian National Health Service. Nonetheless, they may have limited the comparability of our results. 

## 5. Conclusions

The debate is still open on how mandatory vaccination strategies are effective in the short term, but in the long term these strategies may limit confidence in vaccination and lead to an attenuation of the positive effects of coercive measures [22]. Based on these premises, the improvement of vaccination knowledge (health literacy) remains the main target to be achieved, consistent with the current Italian National Vaccination Plan [39]. However, the law on mandatory vaccination for the MMR vaccine in Italy has allowed a significant increase in vaccination coverage rates in both regions analyzed. In the future, a health policy goal should be the improvement of parents’ trust, allowing them to make informed and responsible choices. Specifically, communication and information are essential components of strategies to counteract vaccination hesitation and ensure the success of any immunization program [39,40].

## Figures and Tables

**Figure 1 vaccines-08-00057-f001:**
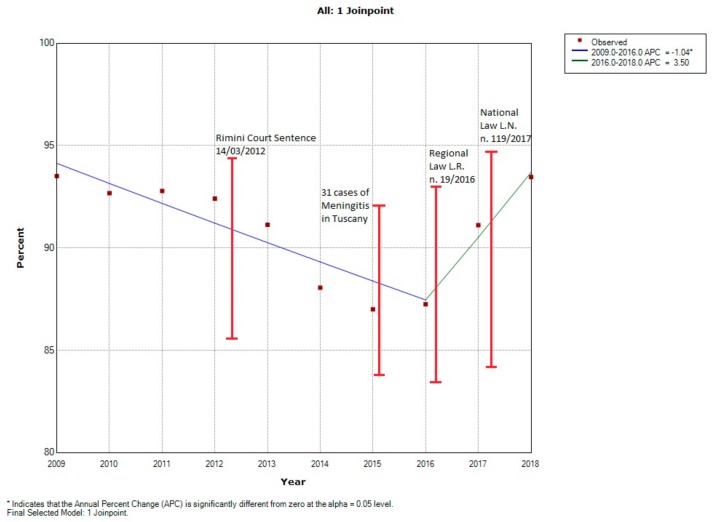
Trend of 24-month vaccination rates per 100,000 inhabitants between 2009 and 2018 in the Emilia-Romagna region.

**Figure 2 vaccines-08-00057-f002:**
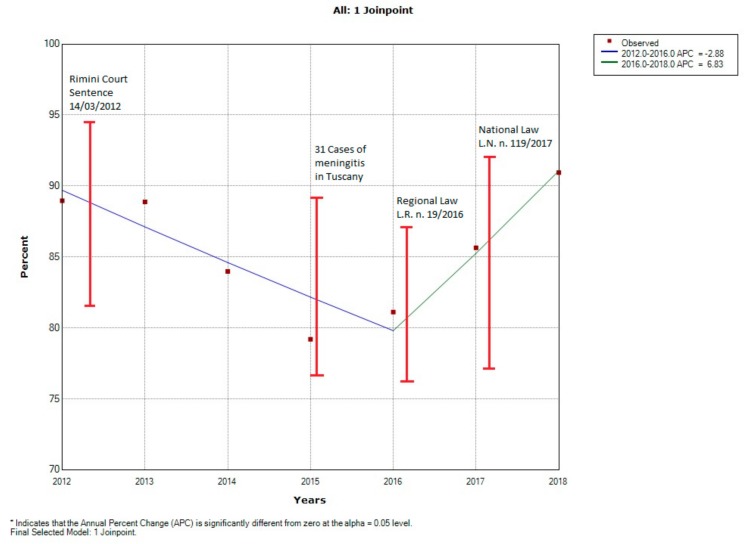
Trend of 24-month vaccination rates per 100,000 inhabitants between 2009 and 2018 in the Sicily region.

**Table 1 vaccines-08-00057-t001:** Vaccine coverage rates at 24 months in the Emilia-Romagna and Sicily regions.

Macro Areas	2009	2010	2011	2012	2013	2014	2015	2016	2017	2018
North Emilia	94.5	94.4	93.8	93.4	92.1	89.3	89.4	89.2	92.4	95.2
Central Emilia	94.0	93.2	93.5	92.6	92.6	89.8	87.4	87.4	91.3	96.1
Romagna	91.1	89.0	90.1	90.4	87.7	83.7	82.1	83.3	88.4	93.6
Western Sicily	88.7	89.9	92.3	91.2	92.2	87.6	82.6	85.4	89.1	92.9
Eastern Sicily	84.1	86.8	88.9	86.9	85.9	80.7	76.1	77.2	82.5	89.8
Emilia-Romagna Region	93.5	92.7	92.8	92.4	91.1	88.1	87.0	87.2	91.1	95.0
Sicily Region	86.8	87.8	91.6	88.9	88.9	84.0	79.2	81.1	85.6	90.9

**Table 2 vaccines-08-00057-t002:** MMR vaccine coverage rates at 7-years-old in Emilia-Romagna and Sicily for children with two doses (full vaccination cycle).

Macro Areas	2009	2010	2011	2012	2013	2014	2015	2016	2017
North Emilia	90.1	90.3	90.4	89.6	91.2	91.1	90.2	88.3	89.6
Central Emilia	90.0	90.0	89.2	90.4	91.2	90.3	89.3	87.9	87.4
Romagna	86.4	87.2	88.4	88.7	90.0	88.8	86.3	86.3	87.5
Western Sicily	N.A.	N.A.	N.A.	N.A.	70.3	72.8	72.2	78.6	90.7
Eastern Sicily	N.A.	N.A.	N.A.	N.A.	58.7	58.9	63.6	69.2	79.2
Emilia-Romagna Region	89.1	89.4	89.6	89.6	90.9	90.3	88.9	87.7	88.5
Sicily Region	N.A.	N.A.	N.A.	N.A.	63.9	65.7	67.6	75.3	84.7
N.A. = No available data									

**Table 3 vaccines-08-00057-t003:** Percentage of vaccinated and unvaccinated children in the Emilia-Romagna region in 2012 and comparisons with the following years.

Year	Unvaccinated (%)	Vaccinated (%)	Total	χ²	*p*-value
2012	3185 (7.6)	38,768 (92.4)	41,953		
2013	3569 (8.9)	36,652 (91.1)	40,221	44.71	<0.001
Total	6754	75,420	82,174		

2012	3185 (7.6)	38,768 (92.4)	41,953		
2014	4700 (11.9)	34,658 (88.1)	39,358	438.77	<0.001
Total	7885	73,426	81,311		

2012	3185 (7.6)	38,768 (92.4)	41,953		
2015	4892 (13)	32,735 (87)	37,627	636.51	<0.001
Total	8077	71,503	79,580		

2012	3185 (7.6)	38,768 (92.4)	41,953		
2016	4716 (12.7)	32,269 (87.3)	36,985	580.88	<0.001
Total	7,901	71,037	78,938		

2012	3,185 (7.6)	38,768 (92.4)	41,953		
2017	3,208 (8.9)	32,874 (91.1)	36,082	43.52	<0.001
Total	6,393	71,642	78,035		

2012	3,185 (7.6)	38,768 (92.4)	41,953		
2018	2,286 (6.5)	32,702 (93.5)	34,988	32.34	<0.001
Total	5,471	68,470	76,941		

**Table 4 vaccines-08-00057-t004:** Percentage of vaccinated and unvaccinated children in the Emilia-Romagna region in 2015 and comparisons with the following years.

Year	Unvaccinated (%)	Vaccinated (%)	Total	χ²	*p*-value
2015	4892 (13)	32,735 (87)	37,627		
2016	4716 (12.7)	32,269 (87.3)	36,985	1.04	0.3077
Total	9608	65,004	74,612		

2015	4892 (13)	32,735 (87)	37,627		
2017	3208 (8.9)	32,874 (91.1)	36,082	318.16	<0.001
Total	8100	65,609	73,709		

2015	4892 (13)	32,735 (87)	37,627		
2018	2286 (6.5)	32,702 (93.5)	34,988	851.35	<0.001
Total	7178	65,437	72,615		

**Table 5 vaccines-08-00057-t005:** Percentage of vaccinated and unvaccinated children in the Sicily region in 2012 and comparisons with the following years.

Year	Unvaccinated	Vaccinated	Total	χ²	*p*-value
2012	5304 (11)	42,733 (89)	48,037		
2013	5201 (11.1)	41,558 (88.9)	46,759	44.71	<0.001
Total	10,505	84,291	94,796		

2012	5304 (11)	42,733 (89)	48,037		
2014	7295 (16)	38,220 (84)	45,515	438.77	<0.001
Total	12,599	80,953	93,552		

2012	5304 (11)	42,733 (89)	48,037		
2015	9167 (20.8)	34,909 (79.2)	44,076	636.51	< 0.001
Total	14,201	77,642	92,113		

2012	5304 (11)	42,733 (89)	48,037		
2016	8296 (18.9)	35,621 (81.1)	43,917	580.88	< 0.001
Total	13,600	78,354	91,954		

2012	5304 (11)	42,733 (89)	48,037		
2017	6204 (14.4)	36,983 (85.6)	43,187	43.52	< 0.001
Total	11,508	79,716	91,224		

2012	5304 (11)	42,733 (89)	48,037		
2018	3752 (9.1)	37,659 (90.9)	41,411	95.92	<0.001
Total	9056	80,392	89,448		

**Table 6 vaccines-08-00057-t006:** Percentage of vaccinated and unvaccinated children in the Sicily region in 2015 and comparisons with the following years.

Year	Unvaccinated	Vaccinated	Total	χ²	*p*-value
2015	9167 (20.8)	34,909 (79.2)	44,076		
2016	8296 (18.9)	35,621 (81.1)	43,917	50.34	0.01
Total	17,463	70,530	87,993		

2015	9167 (20.8)	34,909 (79.2)	44,076		
2017	6204 (14.4)	36,983 (85.6)	43,187	318.16	< 0.001
Total	15,371	71,892	87,263		

2015	9167 (20.8)	34,909 (79.2)	44,076		
2018	3752 (9.1)	37,659 (90.9)	41,411	2,293.1	<0.001
Total	12,919	72,568	85,487

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
