# Peer review of "The Impact of Mandatory Vaccination Law in Italy on MMR Coverage Rates in Two of the Largest Italian Regions (Emilia-Romagna and Sicily): An Effective Strategy to Contrast Vaccine Hesitancy"

_vaccines, 2020, doi:10.3390/vaccines8010057_

Round 1

Reviewer 1 Report

In this paper by Gori and colleagues, the authors analyze the data on measles, mumps, and rubella (MMR) vaccination coverage in two of the largest Italian regions: Emilia-Romagna (RER) and Sicily (SR), between 2009 and 2018, in order to find a possible correlation with index events which could have modified the trend of vaccination rates.

The manuscript provides information and data that will be useful to the scientific community. However, there are some points that require attention before considering the paper for publication in Vaccines.

The English language should be revised in the whole manuscript, possibly with the help of a mother tongue person. There are some typos in the whole manuscript. I would suggest the authors to carefully review and amend it accordingly. As an example, for consistency, authors should refer to Sicilia or Sicily in the whole manuscript. The authors should report published data on MMR vaccination coverage in other Italian regions as well as those at the national level. Do the MMR coverage rates for RER and SR match with the national coverage data?

Author Response

Reviewer 1: Point by point answers

The English language should be revised in the whole manuscript, possibly with the help of a mother tongue person.

We have now revised the English form of the paper.

There are some typos in the whole manuscript. I would suggest the authors to carefully review and amend it accordingly. As an example, for consistency, authors should refer to Sicilia or Sicily in the whole manuscript.

We have checked the whole manuscript and used a consistent terminology.

The authors should report published data on MMR vaccination coverage in other Italian regions as well as those at the national level. Do the MMR coverage rates for RER and SR match with the national coverage data?

Thanks for this comment which allows us to better specify our point and to expand our discussion. We have now added a paragraph and a reference in the discussion section comparing the coverage in our region compared to the coverage at national level, and discussed the differences between northern and southern regions of Italy.

Reviewer 2 Report

Content questions

Is the content relevant? Yes and timely.

Is the information accurate and complete? Please note if there is important information that is missing.

The information is accurate as far as I can tell. 4 references do not lead me to specific content needed. The references need to have more information. See highlights.

Does the information flow in a logical fashion? If not, please suggest an alternative organization plan.

Yes the information flows appropriately.

Are there any key other references that should be added?

See number 2 above and manuscript for needed changes.

Do illustrations, figures, tables, etc. add to the content? Are they clear? Are there others you would recommend?

Yes these are appropriate.

Does the author provide implications?

Yes.

Author Response

Reviewer 2: Point by Point answers

The information is accurate as far as I can tell. 4 references do not lead me to specific content needed. The references need to have more information. See highlights.

Thanks for this comment. We have now improved those references accordingly.

Are there any key other references that should be added? See number 2 above and manuscript for needed changes

We have used the manuscript uploaded in order to improve the highlighted points, and amended the references.

Round 2

Reviewer 2 Report

Authors and Editors

The changes made in this manuscript are appropriate making this  publishable document.